# Effects of Pre-Harvest Supplemental UV-A Light on Growth and Quality of Chinese Kale

**DOI:** 10.3390/molecules27227763

**Published:** 2022-11-11

**Authors:** Youzhi Hu, Xia Li, Xinyang He, Rui He, Yamin Li, Xiaojuan Liu, Houcheng Liu

**Affiliations:** College of Horticulture, South China Agricultural University, Guangzhou 510642, China

**Keywords:** Chinese kale, UV-A light, biomass, antioxidants, glucosinolates

## Abstract

The effects of supplemental UV-A (385 nm) period and UV-A intensity for 5 days before harvest (DBH) on growth, antioxidants, antioxidant capacity, and glucosinolates contents in Chinese kale (*Brassica oleracea* var. *alboglabra* Bailey) were studied in plant factory. In the experiment of the UV-A period, three treatments were designed with 10 W·m^−2^ UV-A supplement, T1(5 DBH), T2 (10 DBH), and no supplemental UV-A as control. In the experiment of UV-A intensity, four treatments were designed with 5 DBH, control (0 W·m^−2^), 5 w (5 W·m^−2^), 10 w (10 W·m^−2^), and 15 w (15 W·m^−2^). The growth light is as follows: 250 μmol·m^−2^·s^−1^; red light: white light = 2:3; photoperiod: 12/12. The growth and quality of Chinese kale were improved by supplemental UV-A LED. The plant height, stem diameter, and biomass of Chinese kale were the highest in the 5 W·m^−2^ treatment for 5 DBH. The contents of chlorophyll a, chlorophyll b, and total chlorophyll were only highly increased by 5 W·m^−2^ UV-A for 5 DBH, while there was no significant difference in the content of carotenoid among all treatments. The contents of soluble sugar and free amino acid were higher only under 10 DBH treatments than in control. The contents of total phenolic and total antioxidant capacity were the highest in 5 W·m^−2^ treatment for 5 DBH. There was a significant positive correlation between total phenolic content and DPPH and FRAP value. After 5 DBH treatments, the percentages and contents of total aliphatic glucosinolates, sinigrin (SIN), gluconapin (GNA), and glucobrassicanapin (GBN) were highly increased, while the percentages and contents of glucobrassicin (GBS), 4-methoxyglucobrassicin (4-MGBS), and Progoitrin (PRO) were significantly decreased, especially under 10 W·m^−2^ treatment. Our results show that UV-A LED supplements could improve the growth and quality of Chinese kale, and 5 W·m^−2^ UV-A LED with 5 DBH might be feasible for Chinese kale growth, and 10 W·m^−2^ UV-A LED with 5 DBH was better for aliphatic glucosinolates accumulation in Chinese kale.

## 1. Introduction

Chinese kale (*Brassica oleracea* var. *alboglabra* L.) originated in South China and spread throughout Southeast Asia thereafter; the flower stalk and the leaf of Chinese kale are the edible organs, which are crisp and abundance in health-promoting phytochemicals such as vitamin C, carotenoids, phenolics, and glucosinolates [1].

Light is an important factor in regulating plant growth, especially in vertical farms with artificial light. Ultraviolet A (UV-A) (320–400 nm) is a chief ingredient of UV in solar radiation. Although UV-A is not photosynthetically active radiation (400–700 nm), as a part of the radiation that could reach the Earth’s surface, it regulates the growth and morphology of the plant. However, there were both beneficial and harmful results about the impacts of UV-A on plant growth and morphology of various species. Adding UV-A to red light had beneficial effects on the growth of tomato seedlings [2,3], while, in tomato fruit, the weight per fruit was significantly lower in the UV-A treatment than in the non-UV-A treatment [4]. Moreover, in pak-choi, under natural light, supplemental both 400 nm and 380 nm UV-A could increase the fresh and dry weight [5]. However, in another study, compared to filter UV-A, non-filter UV-A decreased the fresh and dry weight and leaf area in pak-choi [6]. Moreover, in kale, under different peak wavelengths of UV-A light, the highest growth was observed at 405 nm UV-A compared to 370 and 385 nm UV-A [7]. In Sweet Basil, mild UV-A intensity (10 and 20 W·m^−2^) improved biomass production, while higher UV-A (30 W·m^−2^) intensity reduced biomass production [8]. In Chinese kale, higher biomass was found in 12 h/d UV-A exposure compared to 6 h/d UV-A exposure [9]. Thus, the effect of UV-A on plant growth depends on species, plant growth stage, UV-A wavelength, UV-A intensities, UV-A periods, and methods of supplementing UV-A. 

Plants accumulate a diverse range of secondary metabolites, such as phenolic compounds, vitamin C, and flavonoids, in response to UV radiation [10]. In tomato seedlings, total flavonoids and UV-absorbing compounds were highly increased by UVA radiation [3]. In soybean sprouts, the UV-A treatment increased the content of total ascorbic acid [11]. Compared to supplemental far-red or no supplementary light, significant increases in the contents of vitamin C, flavonoid, polyphenol, and anthocyanin were observed in red lettuce under UV-A [12]. Similar results were observed in the UV-A treatments of Chinese kale baby-leave [13]. Higher contents of total phenolic compounds and total flavonoids were found in broccoli microgreens under UV-A [14]. In kale, supplemental UV-A could stimulate the accumulation of total phenolic [15]. Phenolic compounds, including flavonoids and non-flavonoids, possess high antioxidant activity for human health and plant [16]. Moreover, for aging humans, antioxidant diets could alleviate dementia [17].

Glucosinolates (GLS) are one class of nitrogen- and sulfur-containing secondary metabolites in brassica vegetables and are classified into three classes, including aliphatic, aromatic, and indolic glucosinolates. Intake of glucosinolates and their hydrolysis products (isothiocyanates, nitriles, thiocyanates, and epithionitriles) from brassica vegetables could effectively inhibit many types of cancers and prevent cardiovascular and neurodegenerative disorders [18,19]. In addition, glucosinolates are also responsible for bitter taste, sulfurous aroma, and pungency in brassica vegetables [20].

The increasing awareness regarding the benefits of nutritional food consumption has resulted in growing research seeking better light recipes to produce nutritional vegetables. Previously, the effect of UV-A intensities on Chinese kale baby leaves has been studied [21]. In this study, the effects of different periods and intensities of pre-harvest supplemental 385 nm UV-A LED on marketable mature Chinese kale were investigated to find the optimal UV-A supplement for nutritional vegetable production in plant factories.

## 2. Results

### 2.1. Effects of Different Pre-Harvest Supplemental UV-A LED Periods on the Growth and Quality of Chinese Kale

#### 2.1.1. Growth

The growth of Chinese kale was significantly affected by different supplemental UV-A periods (Table 1). Compared with the control, the stem diameter of Chinese kale in T1 treatment significantly increased by 11.23%, and the fresh and dry weights of the shoots were significantly enhanced by supplemental UV-A treatments, increased by 24.01% and 31.40% in T1 and 27.21% and 28.49% in T2, respectively. However, there was no significant difference in fresh and dry weights of roots between the control and UV-A treatments. 

#### 2.1.2. Photosynthetic Pigments Content

The contents of photosynthetic pigments, including chlorophyll a, chlorophyll b, total chlorophyll, and carotenoid in Chinese kale, were not significantly affected by supplemental UV-A period treatments (Figure 1A). Compared with the control, the ratio of chlorophyll a/b increased by 2.83% under T2 treatment. For the ratio of chlorophyll/carotenoids, there was a 3.42% increase under the T1 treatment but a 3.78% decrease under the T2 treatment (Figure 1B). Significant differences were found in the ratios of chlorophyll a/b and chlorophyll/carotenoids between T1 and T2.

#### 2.1.3. Contents of Soluble Sugars, Free Amino Acids, Nitrate, Total Phenolic and Antioxidant Capacity

No significant differences in soluble sugar content were found among supplemental UV-A treatments and control, while compared with T1, an 18.70% significant increase was seen in T2 (Figure 2A). The effect of supplemental UV-A treatments on free amino acid content was dependent on UV-A radiation days (Figure 2B). Compared with the control, the T2 treatment highly enhanced the free amino acid content of Chinese kale by 38.53%, but no significant difference in the T1 treatment. There was a decreased tendency of nitrate content in Chinese kale treated with supplemental UV-A treatments but only a 13.46% significant reduction in T2 treatment (Figure 2C). The antioxidant capacity in Chinese kale was highly enhanced under different UV-A periods. T1 and T2 treatments significantly induced the accumulation of total phenolic compared with control, by 29.66% and 29.34%, respectively (Figure 2D). Compared with the control, T1 and T2 treatments significantly increased the DPPH value of Chinese kale, with 29.04% and 30.87% enhancements, respectively (Figure 2E). Similarly, FRAP values were significantly higher in T1 and T2 treatments (39.39% and 69.66%) than in the control (Figure 2F), and the most FRAP value was observed in T2. 

#### 2.1.4. Glucosinolates Profile and Content

Nine glucosinolates were identified in both control, and UV-A-treated Chinese kale, including five aliphatic glucosinolates (A-GSL): progoitrin (PRO), glucoraphanin (GRA), sinigrin (SIN), gluconapin (GNA), and glucobrassicanapin (GBN), and fourindolic glucosinolate (I-GSL): 4-hydroxy-glucobrassicin (4-HGBS), glucobrassicin (GBS), 4-methoxyglucobrassicin (4-MGBS), and neoglucobrassicin (NGBS) (Figure 3). 

The total GSLs content was affected by supplemental UV-A period treatments (Figure 3). Compared with the control, the content of total GSLs and A-GSLs were higher in the T1 treatment, with 37.46% and 33.18% increases, but no significant difference was found in the T2 treatment. However, supplemental UV-A irradiance had 7.83% and 9.28% decreases in the total I-GSLs content of Chinese kale under T1 and T2 treatments (Figure 3A).

Contents of individual A-GSL were differently influenced by UV-A supplemental days (Figure 3B). The content of GRA, SIN and GNA increased by 35.42%, 45.94%, and 38.02% in T1 treatment, respectively, compared to the control. However, no significant differences in the contents of PRO and GBN were observed between the control and UV-A treatment. Different UV-A days affected the content of individual I-GSLs in Chinese kale (Figure 3C). Compared with the control, T1 and T2 treatments showed a 31.43% and 20% significant reduction of GBS content in Chinese kale, respectively, but there were no significant differences in other individual I-GSLs. Therefore, T1 treatment was beneficial to the GSLs and A-GSLs accumulation in Chinese kale, whereas supplemental UV-A treatment was not conducive to I-GSLs accumulation.

The percentage of total A-GSLs (90.51–93.47%) in total GSLs was significantly higher than that of total I-GSLs (6.53–9.49%) in Chinese kale (Table 2). Moreover, the highest percentage of GSLs was found in GNA (55.52–57.81%), followed by SIN (21.86–23.88%), but the lowest was detected in NGBS (0.75%–0.96%). Compared to the control, the percentages of total A-GSLs in T1 and T2 treated Chinese kale increased by 3.27% and 1.71% [|T1 or T2 -control|/control], and those in T1 treatment were the highest. However, the percentages of GBS, NGBS, and total I-GSLs significantly decreased under UV-A treatments. The percentage of total I-GSLs in T1 and T2 decreased by 31.19% and 16.33%, respectively. The most reduction (48.19%) of GBS percentage was observed in T1, a 24.35% reduction was found in T2, and a statistical decrease was found in T1 and T2. The percentages of 4-HGNS and 4-MGBS were not significantly influenced by supplemental UV-A treatments.

### 2.2. Effects of Different Pre-Harvest Supplemental UV-A LED Intensities on Growth and Quality of Chinese Kale

#### 2.2.1. Growth

Different supplemental UV-A LED intensities significantly affected the growth of Chinese kale (Table 3). Compared with the control, the plant height and the fresh shoot weight significantly increased by 5 w, 10 w, and 15 w treatments in Chinese kale, which increased by 22.16% and 17.45%, 20.62% and 11.72%, and 14.34% and 11.90%, respectively. The stem diameter of plants was the greatest in the 5 w treatment. The fresh weight root was significantly higher (21.28%) in the 5 w treatment, and there was less difference in the fresh weight of root between plants in 10 w and 15 w compared with those in control. The dry weight of the shoot was highly enhanced by 18.48% in the 5 w treatment, but no significant difference was discovered in the dry weight of the root among UV-A treatments and control.

#### 2.2.2. Photosynthetic Pigments Content

The photosynthetic pigments were affected by different UV-A intensities (Figure 4). Compared with the control, the chlorophyll a content significantly increased by 14.36%; in 5 w, the chlorophyll b content significantly increased by 22.42% and 15.64% in 5 w, and 10 w treatment, respectively, and the total chlorophyll content highly increased by 16.39% in 5 w treatment, while no significant differences were found in the content of carotenoids. There was a 6.49% decrease in chlorophyll a/b in 5 w treatment, but chlorophyll/carotenoids were enhanced by 11.30%, 8.58%, and 1.97% in 5 w, 10 w, and 15 w treatments as compared to the control, respectively.

#### 2.2.3. The Content of Soluble Sugars, Free Amino Acids, Nitrate, Total Phenolic and Antioxidant Capacity

There was a gradually decreased tendency of soluble sugar content in Chinese kale with the increase in UV-A intensity, but only a significant reduction with 19.41% in 15 w treatment (Figure 5A). There was numerically lower content of free amino acids in UV-A treatments than in the control, while no significantly difference was seen in content of free amino acids and nitrate (Figure 5B,C). However, higher contents of total phenolic, DPPH, and FRAP were observed in 5 w, 10 w and 15 w treatments (Figure 5D,F). The total phenolic content increased by 41.16%, 29.13%, and 40.30%, respectively; DPPH increased by 9.16%, 9.25%, and 8.65%, respectively; and FRAP increased by 27.71%, 21.62%, and 25.08%, respectively, in 5w, 10 w, and 15 w treatments.

#### 2.2.4. Glucosinolates Profile

Different supplemental UV-A intensities had different effects on total glucosinolates (GSLs) content in Chinese kale (Figure 6A). Compared with the control, the content of total glucosinolates (GSLs) significantly increased by 5.83% and 10.12%, and the content of total aliphatic glucosinolates (A-GSLs) significantly increased by 13.94% and 19.47% in 5 w and 10 w treatments, respectively, which was the highest in 10 w treatment. However, Chinese kale exposed to UV-A intensities of 5, 10, and 15 w showed a 25.58%, 26.05%, and 27.44% reduction in the total indolyl glucosinolates (I-GSLs) content, respectively. 

The contents of individual A-GSL were differently enhanced by supplemental UV-A treatments, except for progoitrin (PRO) (Figure 6B). In total, 48.36% higher glucoraphanin (GRA) was produced in the 15 w treatment and 34.00% higher glucobrassicanapin (GBN) in the 10 w treatment. The contents of sinigrin (SIN) and gluconapin (GNA) significantly increased both in 5 w (by 13.24% and 19.17%) and 10 w (26.66% and 21.60%) treatments, respectively. However, the PRO content in Chinese kale was reduced by 18.47%, 25.34%, and 37.73% in supplemental UV-A treatments, respectively.

Individual I-GSL contents were significantly affected by different UV-A intensities (Figure 6C). The 4-hydroxy-glucobrassicin (4-HGBS) content of Chinese kale in the 5 w treatment was 32.73% higher than the control. However, the content of glucobrassicin (GBS) and 4-methoxyglucobrassicin (4-MGBS) significantly reduced by 35.78% and 33.94%, 38.97% and 23.21%, and 21.15% and 40.47% in 5 w, 10 w, and 15 w treatments, respectively. There was no significant difference in neoglucobrassicin (NGBS) content between the UV-A treatment and the control.

Supplemental UV-A intensities affected the percentages of A-GSLs and I-GSLs (Table 4). Compared to the control, the percentage of A-GSLs significantly increased by 7.70%, 8.52%, and 6.92% |5 w(10 w,15 w)-control)/control| under 5 w, 10 w and 15 w treatment, respectively, and that in 10 w treatment was the highest. While the percentage of I-GSLs significantly decreased by 29.77%, 32.93%, and 26.75 % under 5 w, 10 w, and 15 w treatment, respectively. Except for PRO, the percentages of other individual A-GSL (GRA, SIN, GNA, GBN) increased under UV-A treatment. The PRO percentages under 5 w, 10 w, and 15 w treatment decreased 23.16%, 32.21%, and 37.27%, respectively, while 12.63%, 10.46%, and 8.92% significant increases in GNA percentage were found under 5 w, 10 w, and 15 w treatment, respectively. The percentage of GRA was significantly higher by 50.63% under the 15 w treatment. The remarkable increase in GBN was only observed under 10 w. No statistical difference was observed in SIN percentages among treatments. For individual I-GSL, the 4-HGBS percentage in the 5 w treatment increased by 25.00%, whereas that in the 10 w and 15 w treatment decreased by 12.05% and 17.41%, respectively. Reductions of 39.37%, 44.48%, and 20.21% were observed in the percentage of GBS, and 37.47%, 30.07%, and 39.98% in the percentage of 4-MGBS in 5 w, 10 w, and 15 w treatment, respectively. No significant difference in the percentages of NGBS was observed among treatments. 

## 3. Discussion

### 3.1. Pre-Harvest Lower UV-A Dosage Was More Beneficial to the Growth of Chinese Kale 

The effects of UV-A on plant growth depend on plant species, intensities, wavelength, and period of UV-A. UV-A might either stimulate or inhibit plant biomass production, even not affect., No pronounced effect was observed on lettuce growth and biomass production when UV-A was supplemented [22]. The shoot fresh weights of dropwort treated for 14 days with 370 nm or 385 nm UV-A LEDs were not significantly different from those of the control [23]. However, remarkable enhancement was seen in plant height, stem diameter, and the fresh and dry weight of 25-day-old Chinese kale baby leaf in 12 h/d UV-A treatment compared with the treatment without UV-A [9]. Moreover, the fresh and dry weights of shoots and roots, and leaf areas greatly increased by UV-A LEDs (370 nm and 385 nm) in kale [15]. Continuous UV-A treatments increased the shoot growth of the sowthistle [24]. The supplemental 385 nm and 400 nm UV-A could increase the biomass of pak-choi [5]. In tomato seedlings, stem diameters, the fresh and dry weights were highly increased in supplemented UV-A treatments until 45 days after sowing, but the growth rate declined under UV-A irradiation after 45 days [2]. In this study, supplemental UV-A positively affected the growth of Chinese kale (Table 1 and Table 3). Shoot weights of Chinese kale were increased in both T1 and T2, while no statistical difference was seen between the two treatments (Table 1). Plants respond to environmental changes by the adjustment of morphology and growth. More biomass accumulates in tomato and radish plants supplemented with UV-A were well correlated with larger leaf size and higher photosynthetic activity induced by UVA [25,26]. In contrast, no significant difference emerged in Chinese kale biomass with the increasing periods of UV-A light, which might be that shorter periods of UVA exposure had induced plant adaptation to the environment. Thus, 5 days before harvest, supplemental UV-A was beneficial for the yield of indoor cultivated Chinese kale, and the highest biomass of Chinese kale was under 5 W·m^−2^ compared with 10 W·m^−2^ and 15 W·m^−2^ (Table 3). The biomass of Chinese kale did not increase with the increasing UVA intensity, which might be a consequence of the plant-saturating response to UVA [27].

Light quality is an important factor in regulating plant growth by affecting chlorophyll and carotenoid syntheses [28].the. Unsuitable UV-A intensity could be a factor for damaging the photosystem II complex [29], and suitable UV-A intensity could be a factor in enhancing electron transport [30]. Various plants respond differently to UV-A. In the leaves of *Rosa hybrida* and *Fuchsia hybrida*, the slight increase in contents of chlorophyll and carotenoids was found under UV-A irradiation [31]. The contents of chlorophyll and carotenoid significantly increased by 3 and 6 days UV-A irradiation in green-leaf lettuce, as well as 3 and 4 days UV-A irradiation in red-leaf lettuce [32,33]. However, there was no difference in the content of chlorophyll and carotenoids in red-leaf lettuce under 6 days of UV-A irradiation [32]. Moreover, no difference in chlorophyll content was found in soybean, under the exclusion of UV-B and UV-B/A components of solar radiation [34]. In this study, there were no obvious differences in the carotenoid contents among UV-A period treatments. However, an increasing trend in chlorophyll content was observed under 5 DBH, and significant increases in the content of chlorophyll a, chlorophyll b, and total chlorophyll were found under 5 W·m^−2^ (5 w) UV-A (Figure 4A), which were similar to the results of Chinese kale under 20 days UV-A with 6 h/d and 12 h/d [9]. Therefore, UV-A irradiation resulted in higher biomass might be due to higher chlorophyll accumulation in Chinese kale [9].

Additionally, the ratios of chlorophyll a/b and chlorophyll/carotenoids were also affected by UV-A. In Chinese kale baby leaves under 10 W·m^−2^ UV-A, the ratio of chlorophyll a/b decreased while the biomass increased [21]. Furthermore, the chlorophyll a/b in Chinese kale showed a decreasing trend with the increase in supplemental UVA exposure duration, while the chlorophyll/carotenoids and biomass showed an increasing trend [9]. Similarly, in this study, the biomass of Chinese kale increased excepted 10 days (T2) UV-A irradiation, the decreasing trend of chlorophyll a/b and the increasing trend of chlorophyll/carotenoids were found in other UV-A (T1, 5 w, 10 w, and 15 w) treatments (Figure 4B). Thus, 10 days UVA treatment might cause stress and protective responses in Chinese kale, while 5 days of treatment might be better for the photosynthetic capacity and the most beneficial UV-A intensity for Chinese kale’s photosynthetic capacity was 5 W·m^−2^.

### 3.2. Pre-Harvest UV-A Effected the Phytochemical Contents of Chinese Kale 

The contents of free amino acids and nitrate in plants reflect the changes in N absorption, transport, and metabolism, which were influenced by light irradiance [35]. Intensive studies have proved UVA exerts an impact on plant biochemistry. The nitrate content was significantly reduced by UV-A with 6 h/d in Chinese kale [9]. Supplemental 12 μmol·m^−2^·s^−1^ UV-A treatment notably decreased nitrate content while increasing proteins content in kale [36]. Moreover, in pak-choi under 380 nm UV-A, decreasing nitrate contents were found, while the increasing trend of soluble protein was observed [5]. Similarly, in this study, with the day extension of the supplemental UV-A, the nitrate content gradually reduced, while the content of free amino acids gradually increased (Figure 2B,C). It might be due to the fact that enzyme activity (NR) related to nitrogen metabolism could be up-regulated by UV-A and nitrate converted to free amino acids [37].

The effect of UV-A on soluble sugars is not conclusive. The reduction of soluble sugar content in green and red pak-choi was induced by 380 nm and 400 nm UV-A [5], while a significant increase was observed in Chinese Kale under 40 μmol·m^−2^·s^−1^ UV-A with 6 h/d [9]. In this study, similar to the trend of the chlorophyll a/b, the soluble sugar contents decreased under UV-A treatments except for 10 DBH UV-A (T2). Generally, low soluble sugar content in plants suggests increased photosynthesis, and high soluble sugar content suggests increased plant resistance [38]. The result of soluble sugar content in this study verified that 10 DBH UVA might cause stress and protective responses in Chinese kale, while 5 days of treatment might be better for the photosynthetic capacity. 

Due to the complex nature of phytochemicals, phenolics groups, DPPH, and FRAP always are employed to evaluate the total antioxidative effects of vegetables. Groups of phenolics act as sunscreens in the leaf epidermis, which protect inner cells from harmful radiation [39]. In lettuce, supplemental UV-A radiation significantly increased the contents of total phenolics and total flavonoids [12]. In tomato fruits, the contents of total phenolics and total flavonoids were highly enhanced under white light supplemented with a UV-A lamp (λmax = 368 nm) [40]. Similarly, in this study, higher total phenolics contents and total antioxidant capacity (DPPH and FRAP) were observed among UV-A treatments. Phenolics as the main component of antioxidants, which is well associated with plant responses to UV-A radiation. The regulatory and structural genes involved in the biosynthesis of the phenolic, such as the phenylalanine lyase (PAL) and chalcone synthase (CHS), could be enhanced by UV-A light [7,41,42,43,44,45]. Phenolic compounds contributed to antioxidant capacity [46]. Therefore, higher total antioxidant capacity (DPPH and FRAP) in Chinese kale under UVA treatment might be highly correlated to the contents of total phenolics. Interestingly, except for FRAP displayed a higher value under 10 DBH UV-A than 5 DBH, no differences were found in the total phenolics and total antioxidant capacity among various UV-A duration and intensities. It indicated that the stimulating effect of UV-A on Chinese kale exhibits a saturation response to the UV-A dose.

Brassica vegetables are rich sources of GSL with high anticarcinogenic activities. In this study, 9 GSLs were quantified in Chinese kale, including 5 A-GSLs and 4 I-GSLs, and A-GSLs (80–93%) were the main GSL, among which GNA was the most predominant one, followed by SIN (Table 2 and Table 4) Previous studies have reported that the contents of A-GSL and total GSL in Chinese kale Baby-leaves remarkably increased with increasing supplementary UV-A intensity, but no difference was observed in I-GSL [21]. Similarly, the content of total GSL and A-GSL increased in Chinese kale baby leaves but decreased in pak-choi under 40 W·m^−2^ 380 nm UV-A [47]. In kale, the content of GSL and I-GSL remarkably increased by 6 and 12 μmol·m^−2^·s^−1^ UVA, while no change was found in 18μmol·m^−2^·s^−1^ UVA treatment [36]. Thus, the effect of UV-A on GSLs biosynthesis varied according to the duration and dose of exposure. In this study, total GSL and total A-GSL contents were promoted by UVA, but these increases in Chinese kale were not linearly increased with the increase in the UVA duration and the UVA intensity (Figure 3 and Figure 6). The stimulating effect might be related to the expression of transcription factors (e.g., DOF1.1, MYB41, MYB28, MYB34) and cure structure genes of glucosinolates (BCATs, MAMs, CYP79s, CYP83, AOPs) and other gene families could be induced by UVA [9,13]. However, this stimulating effect was only observed under short-term exposure to UVA and lower UVA doses, while higher doses of UV-A just slightly increased the content of glucosinolate in Chinese kale in the present study. These might be the consequence of expressions of the above genes varied to different duration and doses of UVA exposure. Therefore, the appropriate UVA light intensity and photoperiod could promote glucosinolate accumulation in Chinese kale, which mechanism still needs further research.

## 4. Materials and Methods

### 4.1. Plant Material and Cultivation Conditions

The experiment was conducted in an artificial lighting plant factory at South China Agricultural University. The seeds of Chinese kale (*Brassica oleracea* var. *alboglabra Bailey*) were sown into soaked sponge blocks and then placed in a dark germination chamber for 2 days. After germination, the seedlings were transferred in deep flow technique (DFT) systems with 1/2 strength Hogland nutrient solution and 250 μmol·m^−2^·s^−1^ white LEDs. At 15 days after sowing, seedlings with 3 true leaves were transplanted into DFT systems with 1/2 strength Hogland nutrient solution; the light and the dark air temperature was 22 °C, with 75% relative humidity, under LED panels (Chenghui Equipment Co., Ltd, Guangzhou, China; 150 cm × 30 cm) with PPFD (photosynthetic photon flux density) of 250 ± 10 μmol·m^−2^·s^−1^ (60% white and 40% red) and 12 h light/dark period (6:00 a.m.–6:00 p.m.).

### 4.2. Supplemental UV-A LED Treatment

For UV-A LED treatment, two different experiments were designed. In experiment 1, supplemental 10 W·m^−2^ UV-A LED (385 ± 10 nm) 4 h per day (6:00–8:00; 16:00–18:00) for 5 (T1) and 10 (T2) days before harvest (DBH) and no supplemental UV-A as control. 

In experiment 2, 5 days before harvest, 0 W·m^−2^ (control), 5 W·m^−2^ (5 w),10 W·m^−2^ (10 w), and 15 W·m^−2^ (15 w) supplemental UV-A LED were used. UV-A LED treatment was supplied for 4 h per day (6:00–8:00; 16:00–18:00) and three biological replicates per treatment, 24 plants per replicate in both experiments. 

### 4.3. Growth Measurements

After supplemental UV-A treatments, the marketable mature Chinese kale plants were harvested. The plant growth parameters, including plant height, stem diameter, and fresh and dry weight, were measured. Plant height was measured from the cotyledon to the highest shoot tip of Chinese kale by rectilinear scale. The diameter of the 4th node from the first true leaf was considered stem diameter. The shoot fresh and dry weight of the product organ (flower stalk above the 4th node was defined as the commercial yield) was measured using an electronic balance. Eight repetitions were conducted. 

### 4.4. Photosynthetic Pigments Content

Four repetitions of fresh leaf tissue were collected for photosynthetic pigments measurement. The chlorophyll content was determined as described earlier [48]. Fresh leaf tissue (0.2 g) was soaked in 8 mL acetone/ethanol (1:1, *v*:*v*) solution in the dark at room temperature until the color faded to white (24 h). Absorbance at 663 nm, 645 nm, and 440 nm was taken by a UV-spectrophotometer (Shimadzu UV-1780, Corporation, Kyoto, Japan). Photosynthetic pigments content was quantified as follows:Chlorophyll a (mg·g^−1^) = (12.7 × OD663 − 2.69 × OD645) × V/W × 1000
Chlorophyll b (mg·g^−1^) = (22.9 × OD645−4.86 × OD663) ×V/W × 1000
Total Chlorophyll (mg·g^−1^) = (8.02 × OD663 + 20.20 × OD645) × V/W × 1000
Carotenoids (mg·g^−1^) = (4.7 × OD440−0.27 × Total Chlorophyll content) × V/W × 1000.
where V is the volume of the extract and W is the weight of the sample.

### 4.5. Contents of Soluble Sugar, Amino Acid, and Nitrate Measurements

In total, 9 randomly selected fresh plants per treatment (3 plant/replicate, 3 replicates/treatment) were placed into liquid N_2_ and stored at −80 °C until analysis. 

Soluble sugar content was determined using anthrone colorimetry [49]. In total, 0.5 g ground sample was mixed with 10 mL distilled water and heated in boiling water for 30 min. Centrifuged the mixture at 3000× *g* rpm for 10 min. In total, 1.9 mL distilled water, 0.5 mL acetate, and 5 mL vitriol were used to mix with 0.1 mL supernatant. The mixture was heated in boiling water for 15 min. 

Amino acid contents were measured using the ninhydrin method with leucine as the standard [50]. In total, 1.0 g ground sample was homogenized in 5 mL 10% acetic acid. The mixture was diluted to 100 mL with distilled water. Then, the shaken well solution was filtered. In total, 1.0 mL supernatant was extracted and mixed with 1.0 mL distilled water, 3.0 mL indenone hydrate, and 0.1 mL ascorbic acid, in turn, shook well. Quickly cooled with cold water and fixed the volume to 20 mL with 60% ethanol. Determined the absorbance at 570 nm by UV-spectrophotometer

The nitrate content was determined by the coloration method of sulfosalicylic acid [51]. In total, a 1.0 g ground sample was homogenized in 10 mL distilled water and heated in boiling water for 30 min. The mixture was cooled for 5 min and filtered. In total, 0.1 mL filtrate was mixed with 0.4 mL 5% salicylic and sulfuric acid, then fully reacted for 20 min, and continue mixed with 9.5 mL 8% NaOH. Determined the absorbance at 410 nm by UV-spectrophotometer.

### 4.6. Antioxidant Content and Antioxidant Activity

The content of total phenolic compounds [52]. In total, 0.5 g ground sample was mixed with 8 mL alcohol for 30 min in ice-bathing. Then centrifuged at 3000× *g* rpm for 15 min, 1 mL supernatant was extracted and mixed with 0.5 mL Folin phenol, 1.5 mL 26.7% Na_2_CO_3_, and 7 mL distilled water, in turn, shook well with each solution added. The absorbance of homogenate was determined at 760 nm by UV-spectrophotometer after 2 h incubation.

The antioxidant activity of the sample was analyzed by using the DPPH method [53]. Three types of mixture were prepared. Ai: 2 mL sample supernatant (the method is the same as above) was mixed with 2 mL 0.2 μmol/L DPPH (dissolved 0.0080 g DPPH in alcohol); Aj: 2 mL sample supernatant was mixed with 2 mL ethanol; Ac: 2 mL 0.2 μmol/L DPPH was mixed with 2 mL ethanol. All Ai, Aj, and Ac were determined absorbance at 517 nm by UV-spectrophotometer. The formula for calculating DPPH radical inhibition percentage was as follows: DPPH (%) = [1 − (Ai − Aj)/Ac] × 100%

FRAP-reducing antioxidant power was assayed with UV-spectrophotometer [54]. In total, 0.4 mL sample supernatant (the method is the same as above) was homogenized with 3.6 mL TPTZ working solution (containing 0.3 mol·L^−1^ acetate buffer, 10 mmol·L^−1^ 2,4,6-tripyridylS-triazine, and 20 mmol·L^−1^ FeCl_3_ at a 10:1:1 ratio (*v*/*v*/*v*)) at 37 °C. The absorbance of the mixture was determined absorbance at 593 nm by UV-spectrophotometer after 10 min.

### 4.7. Extraction and Determination of Glucosinolates

The glucosinolate content of samples was determined using the HPLC method, as described previously, with some modifications [55]. Briefly, glucosinolate was extracted from 0.2 g of the lyophilized sample by adding 4 mL 70% methanol in a 75 °C water bath for 25 min. Then, barium acetate was added, followed by centrifugation. Repeat the extraction once. In total, 5 mL extraction solution and 500 μL 0.5 mg·mL^−1^ sulfatase solution slowly flowed through a homemade column. The reaction was carried out at 28 °C for 12 h and eluted with 2 mL ultra-pure water. The mobile phase A (ultrapure water) and B (acetonitrile) change gradients were: 0–32 min, 0–20% acetonitrile; 33–38 min, 20% acetonitrile; and 39–40 min, 20–100% acetonitrile. The detection wavelength was 229 nm, the flow rate was 1 mL·min^−1^, and the column temperature was 30 °C. Each sample was duplicated in parallel, using 100 μL 5 mg·mL^−1^ 2-propenylglucoside as an internal standard.

### 4.8. Statistical Analysis

Data were shown as means ± standard deviation (SD) of three treatment repetitions. Analyses of variance (ANOVA) followed by Duncan’s multiple range test were conducted using SPSS 17.0 (SPSS Inc., Chicago, IL, USA). The figures were made by Origin 9.0 (Origin Lab, Northampton, MA, USA).

## 5. Conclusions

The growth and the contents of antioxidants (total phenolic) and antioxidant capacity (DPPH and FRAP) of Chinese kale significantly increased when supplemental UV-A LED was applied before harvest. For 5 days before harvest, 5 W·m^−2^ of supplemental UV-A LED might be the most feasible light intensity for Chinese kale growth and antioxidant capacity, while 10 W·m^−2^ UV-A LED was better for aliphatic glucosinolates accumulation in Chinese kale under artificial light plant factories. The molecular mechanisms of UV-A regulating plant growth and nutrient accumulation should be focused on in future research.

## Figures and Tables

**Figure 1 molecules-27-07763-f001:**
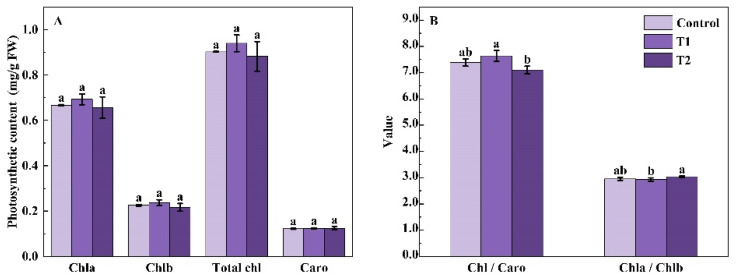
Effects of pre-harvest supplemental UV-A LED period on photosynthetic pigments contents (**A**) and the ratio of photosynthetic pigments in Chinese kale (**B**). The data in the figure represent the means ± the standard deviation error (*n* = 3). Different letters on the bars indicate significant differences at *p* ≤ 0.05 (Duncan’s multiple range test). Chla, Chlb, Total chl, and Caro indicate chlorophyll a, chlorophyll b, total chlorophyll, and carotenoid. Control, T1, and T2 indicate UV-A periods of 0, 5, and 10 days before harvest, respectively, and FW indicates fresh weight.

**Figure 2 molecules-27-07763-f002:**
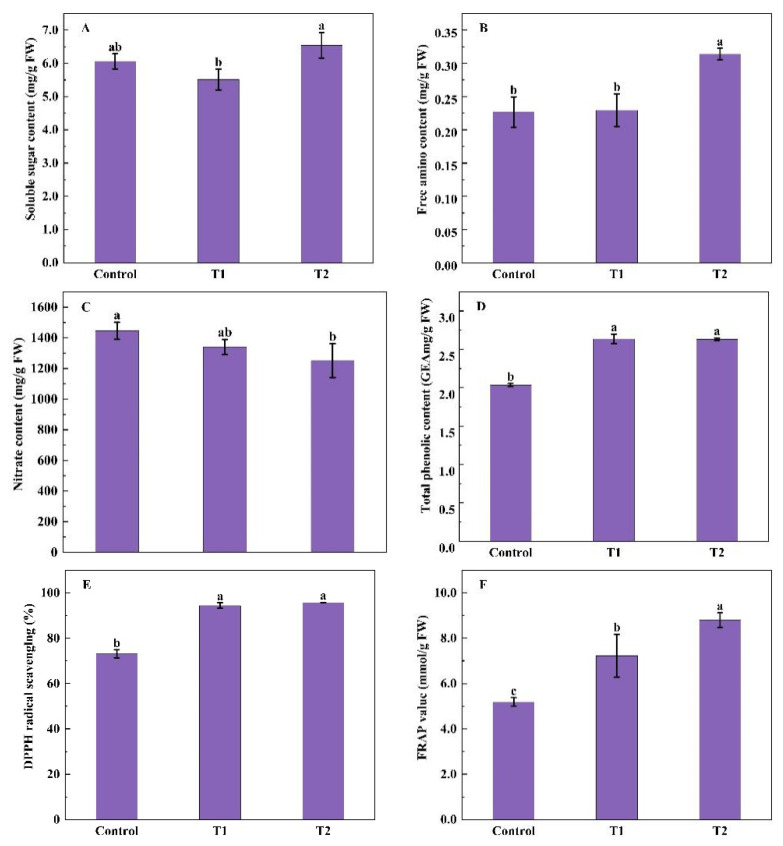
Effects of pre-harvest supplemental UV-A LED period on the content of soluble sugars (**A**), free amino acids (**B**), nitrate (**C**), total phenolic (**D**), DPPH (**E**), and FRAP (**F**) in Chinese kale. The data in the figure represent the means ± the standard deviation (*n* = 3). Different letters on the bars indicate a significant difference at *p* ≤ 0.05 (Duncan’s multiple range test). Control. T1 and T2 indicate UV-A periods of 0, 5, and 10 days before harvest, respectively, and FW indicates fresh weight.

**Figure 3 molecules-27-07763-f003:**
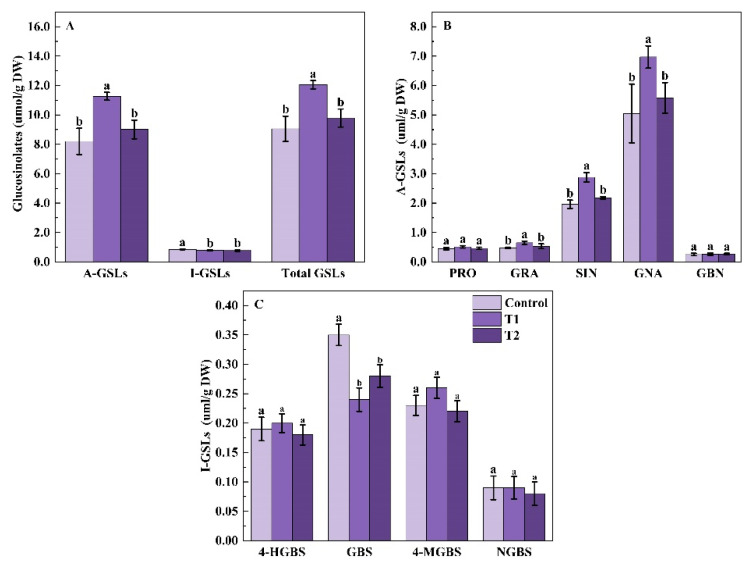
Effects of pre-harvest supplemental UV-A LED period on the contents of total glucosinolates (**A**), individual A-GSLs (**B**), and individual I-GSLs (**C**) in Chinese kale. The data in the figure represent the means ± the standard deviation (*n* = 3). Different letters on the bars indicate a significant difference at *p* ≤ 0.05 (Duncan’s multiple range test). Control. T1 and T2 indicate UV-A periods of 0, 5, and 10 days before harvest, respectively, and DW indicates dry weight.

**Figure 4 molecules-27-07763-f004:**
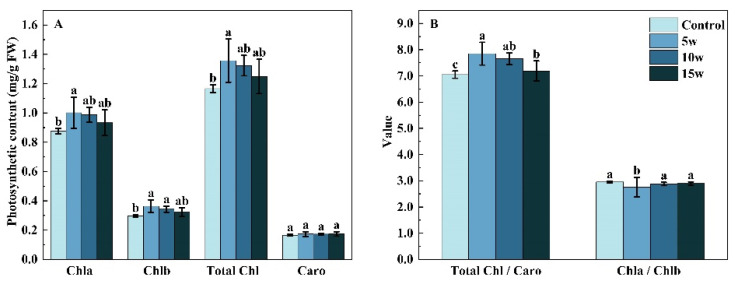
Effects of pre-harvest supplemental UV-A LED intensity on photosynthetic pigments of Chinese kale (**A**) and the ratio of photosynthetic pigments in Chinese kale (**B**). The data in the figure represent the means ± the standard deviation (*n* = 3). Different letters on the bars indicate a significant difference at *p* ≤ 0.05 (Duncan’s multiple range test). Chla, Chlb, Total chl, and Caro indicate chlorophyll a, chlorophyll b, total chlorophyll, and carotenoid. Control, 5 w, 10 w, and 15 w indicate UV-A intensity of 0, 5, 10, and 15 W·m^−2^, respectively, and FW indicates fresh weight.

**Figure 5 molecules-27-07763-f005:**
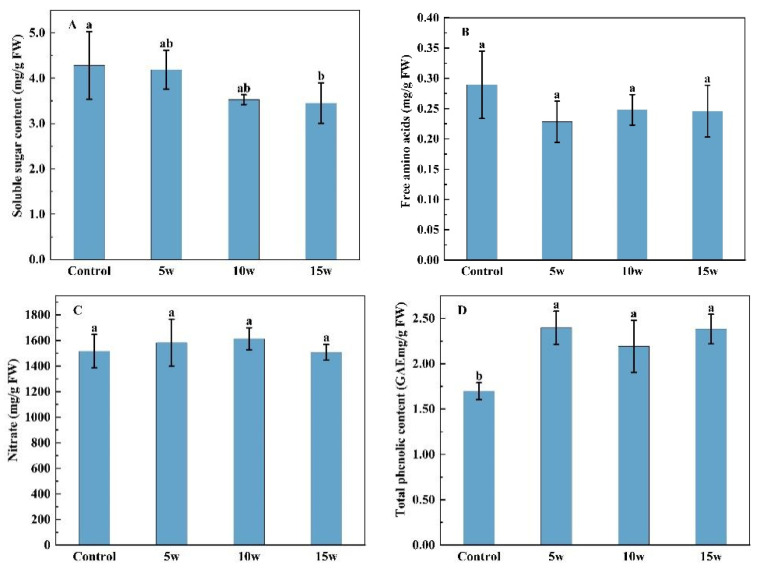
Effects of pre-harvest supplemental UV-A LED intensity on the content of soluble sugars (**A**), free amino acids (**B**), nitrate (**C**), total phenolic (**D**), DPPH (**E**), and FRAP (**F**) in Chinese kale. The data in the figure represent the means ± the standard deviation (*n* = 3). Different letters on the bars indicate a significant difference at *p* ≤ 0.05 (Duncan’s multiple range test). Control, 5 w, 10 w, and 15 w indicate UV-A intensity of 0, 5, 10, and 15 W·m^−2^, respectively, and FW indicates fresh weight.

**Figure 6 molecules-27-07763-f006:**
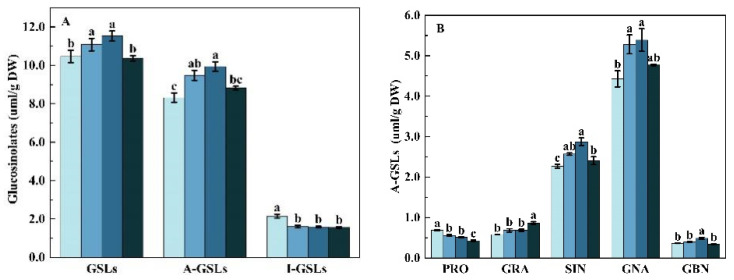
Effects of pre-harvest supplemental UV-A LED intensity on the contents of total glucosinolates (**A**), individual A-GSLs (**B**), and individual I-GSLs (**C**) in Chinese kale. The data in the figure represent the means ± the standard deviation (*n* = 3). Different letters on the bars indicate a significant difference at *p* ≤ 0.05 (Duncan’s multiple range test). Control, 5 w, 10 w, and 15 w indicate UV-A intensity of 0, 5, 10, and 15 W·m^−2^, respectively, and DW indicates dry weight.

**Table 1 molecules-27-07763-t001:** Effects of pre-harvest supplemental UV-A LED period on the growth of Chinese kale.

	Plant Height (cm)	Stem Diameter (mm)	Fresh Weight (g/Plant)		Dry Weight (g/Plant)	
			Shoot	Root	Shoot	Root
Control	19.07 ± 0.76 a	12.38 ± 0.9 b	51.89 ± 6.51 b	5.45 ± 0.42 a	3.44 ± 0.12 b	0.80 ± 0.03 a
T1	19.33 ± 0.9 a	13.77 ± 0.55 a	64.35 ± 4.21 a	5.67 ± 0.75 a	4.52 ± 0.23 a	0.95 ± 0.07 a
T2	19.73 ± 1.32 a	12.99 ± 0.36 ab	66.01 ± 3.88 a	6.20 ± 0.55 a	4.42 ± 0.33 a	0.88 ± 0.09 a

Note: The data in the table represent the means ± the standard deviation (*n* = 3). Different letters in the same column indicate significant differences at *p* ≤ 0.05 (Duncan’s multiple range test). Control, T1, and T2 indicate UV-A periods of 0, 5, and 10 days before harvest, respectively.

**Table 2 molecules-27-07763-t002:** Effects of pre-harvest supplemental UV-A LED period on the percentage of individual glucosinolates in Chinese kale.

Relative Content (%)	Treatments		
	Control	T1	T2
PRO	5.04 ± 0.57 a	4.21 ± 0.36 a	4.67 ± 14 a
GRA	5.26 ± 0.21 a	5.42 ± 0.43 a	5.39 ± 0.52 a
SIN	21.86 ± 3.1 a	23.88 ± 1.54 a	22.28 ± 1.42 a
GNA	55.52 ± 5.21 a	57.81 ± 1.65 a	56.94 ± 1.58 a
GBN	2.83 ± 0.57 a	2.15 ± 0.24 a	2.79 ± 0.12 a
A-GSL	90.51 ± 1.02 c	93.47 ± 0.16 a	92.06 ± 0.73 b
4-HGBS	2.12 ± 0.24 a	1.67 ± 0.11 a	1.87 ± 0.24 a
GBS	3.86 ± 0.5 a	2.00 ± 0.23 c	2.92 ± 0.38 b
4-MGBS	2.55 ± 0.35 a	2.13 ± 0.07 a	2.29 ± 0.19 a
NGBS	0.96 ± 0.12 a	0.75 ± 0.07 b	0.85 ± 0.05 ab
I-GSLs	9.49 ± 1.02 a	6.53 ± 0.16 b	7.94 ± 0.73 b

Note: The content of individual glucosinolates in total glucosinolates was taken as the relative content of individual glucosinolates. The data in the table represent the means ± the standard deviation (*n* = 3). Different letters in the same column indicate a significant difference at *p* ≤ 0.05 (Duncan’s multiple range test). Control, T1, and T2 indicate UV-A periods of 0, 5, and 10 days before harvest, respectively.

**Table 3 molecules-27-07763-t003:** Effects of pre-harvest supplemental UV-A LED intensity on growth of Chinese kale.

Treatment	Plant Height (cm)	Stem Diameter (mm)	Fresh Weight (g/Plant)		Dry Weight (g/Plant)	
			Shoot	Root	Shoot	Root
Control	12.27 ± 0.69 b	11.91 ± 0.47 b	58.81 ± 1.94 b	4.84 ± 0.31 b	3.03 ± 0.28 b	0.67 ± 0.035 a
5 w	14.99 ± 0.47 a	13.00 ± 0.40 a	69.07 ± 2.86 a	5.87 ± 0.17 a	3.59 ± 0.19 a	0.74 ± 0.03 a
10 w	14.80 ± 0.69 a	12.15 ± 0.433 ab	65.70 ± 3.22 a	5.31 ± 0.42 ab	3.13 ± 0.40 b	0.75 ± 0.05 a
15 w	14.03 ± 0.52 a	12.13 ± 0.5 ab	65.81 ± 3.45 a	5.09 ± 0.5 b	3.24 ± 0.36 b	0.74 ± 0.05 a

Note: The data in the table represent the means ± the standard deviation (*n* = 3). Different letters in the same column indicate a significant difference at *p* ≤ 0.05 (Duncan’s multiple range test). Control, 5 w, 10 w, and 15 w indicate UV-A intensity of 0, 5, 10, and 15 W·m^−^^2^, respectively.

**Table 4 molecules-27-07763-t004:** Effects of pre-harvest supplemental UV-A LED intensity on the percentages of individual glucosinolates in Chinese kale.

Relative Content (%)	Treatments			
	Control	5 w	10 w	15 w
PRO	6.52 ± 0.30 a	5.01 ± 0.12 b	4.42 ± 0.31 c	4.09 ± 0.26 c
GRA	5.53 ± 0.31 b	6.15 ± 0.36 b	5.92 ± 0.23 b	8.33 ± 0.61 a
SIN	21.67 ± 0.50 a	23.23 ± 1.58 a	24.99 ± 2.36 a	23.17 ± 1.11 a
GNA	42.27 ± 1.39 b	47.61 ± 1.3 a	46.69 ± 2.41 a	46.04 ± 1.21 a
GBN	3.46 ± 0.14 b	3.55 ± 0.35 b	4.20 ± 0.21 a	3.30 ± 0.10 b
total A-GSL	79.44 ± 0.74 c	85.56 ± 0.38 ab	86.21 ± 0.47 a	84.94 ± 0.28 b
4-HGBS	2.24 ± 0.14 b	2.80 ± 0.19 a	1.97 ± 0.12 c	1.85 ± 0.07 c
GBS	8.61 ± 0.31 a	5.22 ± 0.33 c	4.78 ± 0.33 c	6.87 ± 0.36 b
4-MGBS	8.38 ± 0.31 a	5.24 ± 0.21 b	5.86 ± 0.26 b	5.03 ± 0.66 b
NGBS	1.33 ± 0.14 a	1.18 ± 0.09 a	1.17 ± 0.10 a	1.32 ± 0.05 a
total I-GSL	20.56 ± 0.74 a	14.44 ± 0.38 bc	13.79 ± 0.47 c	15.06 ± 0.28 b

Note: The content of individual glucosinolates in total glucosinolates was taken as the relative content of individual glucosinolates. The data in the table represent the means ± the standard deviation (*n* = 3). Different letters in the same line indicate a significant difference at *p* ≤ 0.05 (Duncan’s multiple range test). Control, 5 w, 10 w, and 15 w indicate UV-A intensity of 0, 5, 10 15 W·m^−2^, respectively.

## Data Availability

Not applicable.

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
