# Peer review of "Effects of Pre-Harvest Supplemental UV-A Light on Growth and Quality of Chinese Kale"

_molecules, 2022, doi:10.3390/molecules27227763_

Round 1

Reviewer 1 Report

General outcomes

The manuscript entitled “Effects of pre-harvest supplemental UV-A light on growth and quality of Chinese kale” presented by Hu et al. that introduce the new trends in application of wavelength, or different energy-related performance, concretely UV-A to agronomical crops - Chinase kale. Also, manuscript offers several very significant quantitative, physiological, of biochemical (antioxidant) parameters. The presented version seems to me logical, well-written, and after “minor revision” I fully recommend to published in Molecules.

Minor comments and suggestion

L36 Here, I suggest adding some information for wide academical auditorium, e.g. For whom, or what does intend for – vertical, precise, or others agriculture-related disciplines? L468 Also, I wish you would place this information in conclusion section.

L35 – 40, and few places in discussion section, please, provide more information about potential mechanisms with directly interaction between the UV-A versus plants responses, you should use more speculative way for better imagination why UV-A was applied?

L44-45 What other factors are beneficial, or related for use UV-A? It is only plant growth, plant species?

L78 T1, and T2 is not fully describe on this place, it poses first stated here, also similar situation is in Tab. 1, please provide the short explanation.

L280 it is correct use vital signal?, L283 Fuchsia hybrida without italic letter? L422 please correct, and control all of used indexes such as L-1.

L288 – 289 Why soybean no difference was observed by using UV-A, UV-B and how to be compare to Chinese kale reactions, species-dependency, please provide some reasonable mechanisms of action.

L306 I do not fully understand the concept of this sentence “This conclusion was validated in terms of biomass.”

L393 please, add dot at the end of sentence.

Reviewer 2 Report

Comments (molecules-1995878)

 The manuscript “Effects of Pre-harvest supplemental UV-A Light on Growth and Quality of Chinese kale” has interesting results and need some modifications before consideration for publication.

Overall introduction section needs improvements. Please provide if there are some earlier studies of using Supplemental light on chines Kale if yes then please mention and if not mention it is first time reported. Moreover, add some data of related species and describe that what the effects on that species are. The data on related species is missing.

Line 53- Moreover, compared with supplemental Far- red, the similar results were observed in UV-A treatment of Chinese kale. Please use the small letter for “far- red light”.

Line 62, 64, 67 use the small letter for “ brassica”

Line 69-  please replace the “functional  vegetables” with “ nutritional vegetables”

Line 282-283. Please use the italic scientific name of Rosa hybrid and Fuchsia hybrid

Line 394- The chlorophyll content was determined by the previous method. Please change the sentence to “The chlorophyll content was determined as described earlier”

Figure 3 please add some more details in legends. Explain the abbreviation of Figs in the legends. Figure legends must be self-explanatory.

Figure 5F please replace the CK with control as all others figures has written the control.  Please add the details of 5w 10 w and 15 w in figure legends.

Please add the abbreviation ( used in the figures) details in Figure 6 

Reviewer 3 Report

The paper “Effects of Pre-harvest supplemental UV-A Light on Growth and Quality of Chinese kale” by authors Youzhi Hu, Xia Li, Xinyang He, Rui He, Yamin Li, Xiaojuan Liu, and Houcheng Liu, submitted to be published in Molecules, represents a detailed investigation of the effects of treatment with three intensities of UV-A (385 nm) for 2 periods to time (5 and 10 days) before harvest on growth, pigment content, level of antioxidants, antioxidant capacity, and glucosinolates contents in Chinese kale (Brassica oleracea var. alboglabra Bailey).

The paper is well structured but needs intensive and serious editing and language improvement.

L104 – “increase” to be removed.

L269 – “treated change” – to be reformulated

L430 – “incubated” to be changed to “incubation”

L380 – “before 5 days of harvest” – to be reformulated.

L47-54, 153-155, 220-221, 262, 269-273, 276-278, 292-295, 298-306, 309-310,315-316 - to be reformulated.

Results

-        Line 90 – an increase with 3.3% cannot be considered significant. Probably the authors mean – statistically different.

-        Table 2 and 7 – it is not indicated how were calculated the presented results – what values were taken as 100%.

-        The quality of figures 1, 3, 4, 6 to be improved. The text on x ordinate and the letters for statistics are not well visible. Part of figure caption in Fig. 3 is in italic, to be corrected.

-        Presenting the results in figure 6 should be improved.

Discussion

-        In general. it is just repletion of description of results, not a real discussion is given. To be corrected accordingly.

Materials and Methods

-        It is indicated that only 2 experiments were performed: Ex. 1 – for determination of treatment with UV-A (10 W m-2) for 5 (T1) and 10 days (T2) before harvest and Ex. 2 – treatment with UV-A – 0, 5, 10 and 15 W m-2 for 5 days before harvest. So only 1 ex. Was performed for the 2 experimental set up was performed, with 3 biological replica. In this respect it is not correct in the figures and table to give SE, it is better to present the SD.

-        It is not indicated the temperature at which the pigment extract was performed. It has to be indicated that the presented values were calculated per g FW.

-        In addition, to be indicated from what article were taken the formulas for calculation of pigments.

-        For the growth conditions – it is not indicated the temperature during the dark period.

In general, the manuscript needs major revision.

Round 2

Reviewer 3 Report

The revised version of the manuscript was improved but there are still issues that were not addressed.

In the response of authors is indicated that in lines 166-167 and 262-263 is given explanation for the way of calculation of percent’s of presented results in table 2 and 4. Unfortunately, in the indicated lines there is no description how the percent was calculated and what values were taken as 100%. It is difficult to follow what represent the listed values – it is activity that was left after UV-A treatment or was decreased with the indicated percent. For Table 2 the given percent for GBS does not correspond to what is mentioned in the text in line 159-160. The whole paragraph (line 152-161) to be checked and corrected accordingly. For Table 4, the whole paragraph line 243-257, the description of results does not correspond to the presented results in the table. To be checked carefully and to be corrected accordingly.

The text on x and y ordinates in Figures 1, 3, 4 and 6 are still with low quality – to be improved accordingly.

For Table 3 – in the table states “Fresh weight” but in the caption of the Table is given as Abbreviation “FW”. There is no need of this abbreviation line 182.

Figure 6 – to be checked what values in what panel is presented “224-235”.

Line 280-282 – needs revision. The higher leaf size does not indicate higher photosynthetic activity.

Line 283-285 – not clear sentence. To be corrected.

All added lines in Discussion need English language correction (279-289, 299-304).

Line 391-292 – how is that – UV-A is damaging PSII and at the same time increasing electron transport?

Line 303-304 – higher biomass can contribute to higher chlorophyll content but is not directly connected with increase of electron transport.

Line 311-314, 318-320, 327-333, 339-349, 358-369 – still need revision.

The whole text to be checked again for English language, still needs improvement.

The manuscript still needs major revision.
